# Genetic Regulation of Mitosis–Meiosis Fate Decision in Plants: Is Callose an Oversighted Polysaccharide in These Processes?

**DOI:** 10.3390/plants12101936

**Published:** 2023-05-09

**Authors:** Harsha Somashekar, Ken-Ichi Nonomura

**Affiliations:** 1Plant Cytogenetics Laboratory, Department of Gene Function and Phenomics, National Institute of Genetics, Mishima 411-8540, Japan; harsha@nig.ac.jp; 2Department of Genetics, School of Life Science, The Graduate University for Advanced Studies (SOKENDAI), Mishima 411-8540, Japan

**Keywords:** anther development, callose, cell cycle, mitosis–meiosis fate decision, meiosis initiation, plant reproduction

## Abstract

Timely progression of the meiotic cell cycle and synchronized establishment of male meiosis in anthers are key to ascertaining plant fertility. With the discovery of novel regulators of the plant cell cycle, the mechanisms underlying meiosis initiation and progression appear to be more complex than previously thought, requiring the conjunctive action of cyclins, cyclin-dependent kinases, transcription factors, protein–protein interactions, and several signaling components. Broadly, cell cycle regulators can be classified into two categories in plants based on the nature of their mutational effects: (1) those that completely arrest cell cycle progression; and (2) those that affect the timing (delay or accelerate) or synchrony of cell cycle progression but somehow complete the division process. Especially the latter effects reflect evasion or obstruction of major steps in the meiosis but have sometimes been overlooked due to their subtle phenotypes. In addition to meiotic regulators, very few signaling compounds have been discovered in plants to date. In this review, we discuss the current state of knowledge about genetic mechanisms to enter the meiotic processes, referred to as the mitosis-meiosis fate decision, as well as the importance of callose (*β*-1,3 glucan), which has been unsung for a long time in male meiosis in plants.

## 1. Introduction

Meiosis is a special type of cell division in eukaryotes whereby one round of DNA replication followed by two consecutive cell divisions produces four haploid gametes, or, in land plants, spores. Though meiosis is believed to have evolved from mitosis, it differs from mitosis in many aspects, such as programmed DNA double strand break (DSB) formation, meiosis-specific chromosome assembly (homolog alignment and synapsis), crossing over, and reductional chromosome segregation. Consequently, these events make the meiosis division process intricately long. Many of the characteristic meiotic events take place in the prolonged G2 phase, or prophase I. Chromatin loading of meiotic proteins/factors coincides with premeiotic DNA replication as a prerequisite to faithfully execute the complicated meiotic processes in sexually reproducing organisms such as yeasts [1,2], plants [3], and mammals [4]. Thus, studies on the meiosis-specific mode of cell cycle control (hereafter we call it “meiotic cell-cycle control”) and related aspects around the premeiotic DNA synthesis phase (S phase) are important to understand the molecular machinery that drives meiosis initiation and progression.

While the major meiotic processes, such as pairing and crossover formation, are substantially conserved, the systems to establish mitosis–meiosis fate decisions greatly differ among species. In both budding and fission yeasts, the starvation of nitrogen and/or carbon sources is required for the induction and initiation of meiosis and to produce stress-tolerant spores [5,6]. In budding yeast, nutritional starvation activates *Initiator of Meiosis1* (*IME1*), a master regulator of meiotic genes [7,8]. However, fission yeast does not conserve IME1; instead, the starvation signal induces the inactivation of Pat1 kinase, a negative regulator of meiosis [9,10]. In *Chlamydomonas reinhardtii*, a unicellular green soil alga, nitrogen starvation triggers fertilization of *plus* and *minus* gametes, and the zygote resistant to freezing and desiccation starts meiosis when ambient conditions improve [11]. Multicellular organisms achieve meiosis through highly coordinated systems involving both cell-autonomous and non-autonomous mechanisms. In mice, retinoic acid acts paracrinely in the differentiation of germline cells and meiosis initiation, while the timing of meiotic entry differs between the sexes [12,13,14]. Only in the ovary has the role of retinoic acid in meiosis been demonstrated in humans [15,16].

In flowering plants, the archesporial cell (ARC), a precursor of spore mother cells (meiocytes) and somatic cell layers surrounding meiocytes, differentiates at the hypodermis of stamens and ovule founders, shortly after completion of floral organ development and termination of the floral meristem [17,18]. In the stamen, ARCs divide periclinally to produce sporogenous cells (SPCs) and undifferentiated primary parietal cells as wrapping SPCs. After several mitotic division cycles, SPCs mature into male meiocytes or pollen mother cells (PMCs) to produce haploid microspores via meiosis. Microspores eventually produce tricellular pollen via two rounds of mitotic division and are supplied for fertilization. Parietal cells further divide periclinally and produce several somatic-cell layers. Eventually, the ARC lineage comprises the pollen sac, or microsporangium, with a concentric whorl structure of SPCs, the tapetal cell (TC) layer, the middle layer, and the endothecium, from inside to outside, within an epidermal layer [19] (Figure 1). The molecular mechanisms underlying floral development, central meiotic events such as homolog pairing and crossing over, and post-meiotic gametogenesis have been well studied and extensively reviewed [20,21,22,23,24,25]. In contrast, much has remained to be elucidated regarding the genetic and epigenetic mechanisms driving the processes from ARC initiation to meiosis in flowering plants.

In this review, we have tried to fill in a missing link in the mitosis–meiosis fate decision in flowering plants. In the later part, we extensively review the current knowledge about callose deposition in premeiotic anthers as a plant-specific system and discuss its relationship with the initiation and progression of male meiosis.

## 2. Meiotic Cell-Fate Decision in Flowering Plants

Initiation, development, and differentiation of the inflorescence and floral meristem have important consequences for the reproductive success and fitness of plants. Changes in environmental conditions such as day length and temperature trigger the reproductive phase transition in flowering plants [26]. The expression of AGAMOUS, a MADS box transcription factor, is coordinated in response to the environment by an unknown mechanism to regulate the identity of the floral meristem and the formation of floral organs [27,28,29,30]. After the floral meristem terminates, the primordial germ cells start dividing asynchronously and encounter a series of fate decisions until they mature into the spore mother cells determined to undergo meiosis. Here we refer to such developmental and cellular fate decisions in young florets concerning meiosis as “meiotic cell-fate decision”. This process consists of multiple and complicated steps during floral development, intervening between reproductive transition and meiosis, and makes it unclear whether the environmentally-triggered meiosis initiation operating in unicellular organisms is also conserved in multicellular plants or not.

Nevertheless, pioneering research in maize revealed that germ cell fate requires a hypoxic microenvironment at the central anther lobes, whereby the replacement of oxygen with nitrogen in tassel-bearing stems resulted in a significant increase of ARCs in stamens [31]. The rice *MEIOSIS ARRESTED AT LEPTOTENE1* (*MEL1*) gene, which encodes an argonaute protein, expresses in ARCs, SPCs, and spore mother cells. In the *mel1* mutant, though meiocytes can enter meiosis, meiotic chromosome condensation seems to be stopped around leptotene [32]. MEL1 associates with 21-nucleotide phased small interfering RNAs (21nt phasiRNAs) [33] and is involved in massive reprogramming of gene expression in anthers, probably for faithful meiosis progression [34,35], suggesting the importance of small RNA-mediated pathways in the meiotic cell-fate decision. Rice MICROSPOROCYTELESS1 (MIL1) is a plant-specific CC-type glutaredoxin required for switching the SPCs to meiosis, and in the *mil1* mutant, the anther locule is abnormally filled with somatic cells [36]. The loss of function of a rice gene encoding ELECTRON TRANSFER FLAVOPROTEIN SUBUNIT β (ETFβ) displays defects in SPC development and meiosis initiation in nitrogen-poor conditions, and the application of excess nitrogen to the *etfβ* mutants completely repaired meiotic defects and sterility [37], thus underscoring the role of micronutrients in meiotic cell cycle control.

In *Arabidopsis*, the centripetal distribution of auxin, a phytohormone, guides the germ cell specification in anthers by *SPOROCYTELESSNESS/NOZZLE* (*SPL*), which encodes the MADS box transcription factor for the differentiation of male and female SPCs [38,39]. Similarly, rice SPL is required for meiotic fate acquisition by SPCs [40]. In SPL-overexpressed plants, the expression of *YUCCA2* (*YUC2*) and *YUC6* genes, both encoding proteins key for auxin biosynthesis, was down-regulated [41], suggesting the repression of auxin-responsive pathways may coincide with differentiating SPCs acquiring a meiotic-cell fate. In contrast, during early floral organogenesis, the auxin biosynthetic pathway genes, such as *YUC4*, are kept activated by AGAMOUS to direct the floral stem cell fate [42]. Thus, there seems to be a strict regulatory network oscillating the auxin levels spatio-temporally for several fate decisions in young anthers. Under high temperature conditions, endogenous auxin levels decrease in the anthers of barley and *Arabidopsis*, leading to reduced male sterility, but externally applied auxin completely recovers fertility [43]. This also implies the importance of auxin in germ cell development, although it is yet unclear in which developmental processes it plays a role.

## 3. Meiotic Cell Cycle Control in Flowering Plants

In many species of flowering plants, prior to the onset of meiosis, the SPCs proliferating asynchronously must terminate their mitotic mode of cell cycle and fully mature into PMCs. The PMCs within anther locules coordinate to establish synchrony and enter meiosis faithfully. Synchronous and/or timely initiation and progression of male meiosis, tightly coupled with strict cell cycle controls, is vital for pollen viability and fertility [44,45]. In sexual organisms, the most common factor involved in both mitosis and meiosis cell-cycle control is the CDKs/cyclin complex, although their mechanism is organism-specific. In yeasts and mammals, simultaneous activation and inactivation of CDC25 phosphatase and Wee1 kinase, respectively, is key for mitosis cell-cycle initiation after DNA replication [1,46,47]. CDC25 activates the CDC2/Cyclin-B complex, the mitosis promoting factor (MPF) required for orderly progression of the G2/M transition and cell-cycle checkpoints [48,49]. Plant cell-cycle progression is regulated by orderly activated CDK/cyclin complexes at different cell-cycle phases [50,51]. In *Arabidopsis*, TARDY ASYNCHRONOUS MEIOSIS I (TAM)/CYCA1;2, an A-type cyclin, is essential for synchronous and timely progression of meiosis [52,53,54], and SOLO DANCERS (SDS) is a plant-specific cyclin involved in both male and female meiosis progression [55]. In *Arabidopsis*, after CDKA1;1-dependent repression of RETINOBLASTOMA-RELATED PROTEIN1 (RBR1) is released by the CDK inhibitors KIP-RELATED KINASES (KRPs), active RBR1 represses the homeodomain transcriptional factor WUSHEL (WUS), a key regulator of stem cell fate in plants, which directs megaspore mother cells to meiosis [56]. In maize, the cell cycle switch to meiosis is mediated by AMEIOTIC1 (AM1), a coiled-coil protein, and the *am1* mutation results in the replacement of meiosis by mitosis-like division in both male and female meiocytes [57,58]. However, the mutation in *DYAD/SWI1*, the *Arabidopsis AM1* homolog, shows defects in recombination, sister chromatid cohesion, and bivalent formation in male meiocytes; female meiosis is replaced by mitosis such as in the maize *am1* mutant [59,60]. Interestingly, *Arabidopsis swi1/dyad* mutants can set seeds when pollinated with wild-type pollen, although such seeds were found to be triploids owing to unreduced female gametes due to mitosis-like division [61]. The rice *am1* mutant displays PMCs arrested at the meiotic leptotene/zygotene transition, different from maize and *Arabidopsis* mutants [62], pointing out a species-specific functional divergence of AM1 in meiotic cell cycle control. The precise mechanism of AM1 meiotic control is still ambiguous; however, it may be required for the timely expression of meiotic genes in both meiocytes and the tapetum [63].

Transcriptional and post-transcriptional regulation of meiotic cell cycle progression and synchrony establishment has also been uncovered in plants. *Arabidopsis* DUET is a PHD finger protein putatively involved in transcription regulation and chromosome organization in male meiocytes, and its loss-of-function mutant exhibits delayed prophase I and prolonged metaphase, I.; resulting in dyads in place of tetrads [64]. *Arabidopsis* FEHLSTART (FST) is another plant-specific bHLH transcription factor involved in the establishment of synchronous and timely entry into male meiosis [65]. Similarly, in petunia, the MEIOSIS-ASSOCIATED ZINC-FINGER PROTEIN1 (MEZ1) transcription factor is also required for male meiosis progression and synchrony establishment [66]. In rice, MEIOSIS ARRESTED AT LEPTOTENE2 (MEL2), an RNA recognition motif-containing protein, is a putative post-transcriptional or translational regulator for proper timing of the mitosis–meiosis transition, and in the *mel2* mutant, PMCs enter meiosis asynchronously, with a subset continuing the mitotic cycle [67]. LEPTOTENE1 (LEPTO1), a type-B response regulator, is required for the leptotene/zygotene transition in rice [68]. Studies have also established a link between histone variants and meiosis synchrony [69], though it is unclear whether histone governance on synchrony is direct or indirect.

Delay or acceleration in the meiotic cycle as observed in the above examples can be attributable to perturbations in the internal pace of meiotic events, leading to the activation of checkpoints associated with cell cycle control at various stage points. For instance, inactivation of *Arabidopsis* MutL HOMOLOG3 (MLH3), a homologue of prokaryotic MutL mismatch repair protein, reduces the crossover numbers by 60%, and as a consequence, prophase I is significantly delayed by about 25 h [70] (see summarized Table 1).

## 4. Callose: A Hallmark for Meiosis Initiation in Flowering Plants

Transition to meiosis in flowering plants entails extensive remodeling of germ cell walls, whereby the cellulosic PMC walls are drastically rearranged and replaced by cell wall *β*-1,3-glucan polymer, called callose [74]. In anthers, SPCs proliferate mitotically and asynchronously, and a typical callose deposition is visible only at the dividing cell plates of SPCs (Figure 2). During the pre-meiotic interphase, i.e., just before meiosis entry, pre-existing cellulose in mature SPC/PMC walls begins to disappear gradually. In parallel to this, a considerable amount of callose is secreted from PMCs to completely fill the extracellular spaces of anther locules and give a “callosic barrier” appearance in the central locule (Figure 2 and Figure 3). In fact, some primitive studies have suggested that callose deposits in meiotic anthers act as a barrier or “molecular filter” to transmit signals important for meiosis, in addition to mechanical isolation of meiocytes from surrounding somatic tissues of anthers [75,76] and ovaries [77,78], although later studies have called into question the impermeable nature of callose [79,80]. Other research have suggested that callose functions as a protective layer for developing sporocytes from the influence of surrounding somatic cells [81,82] and that it helps in preventing the fusion of PMCs during meiosis and aids in the release of microspores from tetrads [83].

A considerable amount of knowledge about callose roles in reproductive parts has come from extensive research dedicated to cytokinesis, tetrads, pollen development, pollen tube germination, and ovule development [84,85,86,87]. In contrast, despite the remarkable callose accumulation having long been thought of as a “histological hallmark” of male meiosis initiation [75], its biological function and relation to the mitosis-to-meiosis transition have attracted little attention for a long time. One of the reasons is that most of the so-far reported mutants and plants that lack callose deposition were reported to normally pass through the meiosis cycle [58,84,85,87,88,89]. On the other hand, we suppose that observations of normal meiosis in callose-lacking mutants may be due to an oversight attributable to the authors’ low interest in meiosis, as we mention below.

The maize *am1* and *mac1* mutants, accompanied by meiotic cycle impairment, have exhibited unusual callose deposition atypical of grasses in both anthers and ovules [85,89]. In the rice *mel2* mutant, the timing of mitosis–meiosis transition is severely affected [67], concomitantly with the complete lack of premeiotic callose deposition to anther locules [73]. In the rice *lepto1* mutant, meiotic chromosome organization is affected, and expression levels of callose metabolism-related genes, *GLUCAN SYNTHASE-LIKE5* (*OsGSL5*) and *UDP-GLUCOSE PYROPHOSPHORYLASE1* (*UGP1*), were significantly reduced [68]. It is also possible that the callose metabolism may be coupled to the early fate decisions in anthers, as underscored by rice *SPOROCYTELESS* (*OsSPL*), whose function is required for SPC differentiation and meiotic fate acquisition, and the loss of OsSPL results in complete loss of PMCs and callose deposition [40]. Owing to fewer attempts and a limited focus on meiosis, the biological reationship of callose with the mitosis–meiosis transition and meiotic cycle progression is currently very limited.

## 5. OsGSL5, a Central Player in Callose Biosynthesis in Anthers at Premeiosis and Meiosis

Recently, our group unveiled the important role of callose in regulating the timely initiation and progression of male meiosis in rice anthers [73]. OsGSL5 is responsible for fulfilling the intercellular spaces of locules in the anther undergoing premeiosis and meiosis (hereafter we call them premeiotic and meiotic anthers). In *Osgsl5* mutant anthers, premeiotic and meiotic calloses were largely depleted. Consequently, premeiotic DNA replication and meiosis progression occurred precociously, and the loading of ZEP1, a transverse filament component of the synaptonemal complex in rice, onto meiotic chromosomes was depleted, accompanied by chromosome anomalies in not all but a subset of PMCs [73]. However, at the end of meiosis, anther locules are largely occupied with normal tetrads, probably due to the abortion of PMCs retaining aberrant chromosomes, while subsequent pollen development is severely affected due to a lack of OsGSL5-dependent calloses surrounding tetrads, as Shi et al. (2014) reported [72]. The frequent appearance of normal tetrads and microspores in callose-deficient mutants probably has long misled researchers into believing premeiotic callose deposition is dispensable for meiosis.

In premeiotic interphase anthers, callose fills the extracellular spaces as encasing PMCs, while OsGSL5 specifically localizes at the plasma membranes (PMs), where PMCs face each other (Figure 2, Panel C). Such a distribution pattern observed in premeiotic and early meiotic anthers argues that the PMC-PMC junction functions as the callose biosynthetic center and further indicates that PMCs have some polarity defined by the heterogeneous distribution of OsGSL5 and callose during premeiosis and meiosis, although the biological meaning of this polarity is enigmatic. Such an uneven distribution of PM-anchored GSL proteins seen in premeiotic anther locules is also found at the tip of elongating pollen tubes (PTs), where the secretary vesicles concentrated at the tip correspond to callose and GSL accumulation [90,91,92,93]. Furthermore, actin filaments, highly concentrated at PT tips, termed “actin fringe”, move along the calcium gradient and synergistically control the cell polarity of PTs [94,95]. In fact, Ge et al. (2009) have reported increased calcium precipitates in the callose walls during meiosis onset in tobacco anthers [96]. Thus, it is possible that similar actin fringe-like organization and/or Golgi-derived vesicle concentration due to increased calcium concentration in premeiotic anthers may be driving the GSL5 polarity (Figure 4). Exocysts are also suggested to be carriers of callose synthases [97], though this idea has yet to be evidenced in anthers.

GSL-dependent callose biosynthesis relies on UDP-glucose (UDP-G) as a substrate. There are many studies on callose synthesis and deposition in anther development and male sterility, and a presumable model in premeiotic and meiotic anthers is summarized in Figure 4. UDP-G is synthesized from sucrose by a coordinated reaction involving at least four enzymes: invertase (INV), hexose kinase (HXK), sucrose synthase (SuSy), and UDP-G pyrophosphorylase (UGPase) (Figure 4). After being unloaded from phloem connective tissue [98], sucrose is cleaved by INV in the apoplast [99,100], or by SuSy in the cytoplasm [90,101]. The resultant hexose monomers are phosphorylated to glucose-1-phosphate (G-1P) in the cytoplasm by HXKs [102]. UGPase interacts with membrane-bound dimerized lectin-rich receptor kinase (LecRK) and converts G-1P and UTP to UDP-G and pyrophosphate [87,103]. The UDP-G supplied via the INV-LecRK-UGPase pathway and/or the SuSy-LecRK-UGPase pathway is then transferred to plasma-membrane-anchored GSL for callose synthesis by UDP-glucose transferase (UGT1) that interacts with GSL/Cal and GTP-bound ROP1 to form a functional callose synthase complex [104] (Figure 4). The *β*-1,3-glucans are proposed to have gel-like properties in a pH dependent manner, whereby an alkaline condition can induce callose gel-formation, whereas an acidic environment makes glucans a rigid polymer [105,106,107]. Local regulation of apoplastic pH by vesicles, together with the cellulose concentration, could influence callose physical properties and drive its movement within apoplast space [108,109,110] (Figure 4).

Previously proposed roles of callose in anthers, such as mechanical support, prevention of PMC cohesion, or as a protective layer of PMCs against surrounding tissue [77,78,79,80,81,82], are insufficient to explain the impact of callose on meiosis-specific events such as precocious meiosis initiation and progression, defective synapses between homologues, and defects in meiosis-specific chromosome condensation/behavior [73]. However, it may be worth considering the role of callose in cell–cell communication via the regulation of PD (symplastic pathway) [111] and membrane permeability (apoplastic pathway) [80,108]. It is known that meiotic PMCs are interconnected with each other and with surrounding TCs through cytoplasmic channels (CCs) believed to have derived from severed PD as meiocytes progress into meiosis. CCs are thought to be important for establishing synchrony between male meiocytes in anther locules [75,76,112,113,114], while later studies demonstrated that CCs cannot solely determine synchrony [115], suggesting the regulation of meiosis synchrony and/or timing in PMCs, tightly coupled with control of premeiotic S phase, may involve other signals operating in conjunction with CCs and cell cycle-controlling proteins. Considering the number of PDs or CCs connecting anther locular cells decreases as meiosis progresses [112], it is possible that callose may contribute to switching the mode of cell-cell communication from symplast to apoplast paths for proper meiosis transition and progression. More studies may be necessary to illustrate the functionality of symplast/apoplast pathways in relation to the meiotic cycle.

OsGSL5 conserves all three domains characteristic of GSL family proteins: Vta1, FKS, and glucan synthase, while several rice GSLs lack the Vta1 domain (Figure 5). We deemed the OsGSL5 protein, which has all three domains, a full protein and extracted GSLs/CalSs, conserving the three typical domains and showing the highest amino-acid sequence identity with OsGSL5 in other plant groups to study their phylogenetic relationship. The analyses revealed that OsGSL5-like proteins are conserved across all land plants, and those in bryophytes made the most ancient clade in the phylogenetic tree (Figure 6). This indicates that OsGSL5-like proteins may have appeared only in land plants after their divergence from charophytes and that the Vta1 domain, together with the FKS domain, may have contributed to protein complexities in land plant evolution.

## 6. Fate of Meiosis and Pollen Formation in Plants Lacking Meiotic Callose

Certain plant species with PMCs lacking callose are able to initiate and complete meiosis. For example, *Pandanus odoratissimus* and *Pergularia daemia* naturally lack callose walls but can still complete the meiosis cycle [120,121]. Despite the completion of meiosis in the absence of callose, meiocytes show monads instead of dyads after meiosis, I.; likely because of the absence of callose. At the end of meiosis, I.I.; microspores are normally formed in *Pandanus odoratissimus* [120], but the exine formation on pollen is impaired in *Pergularia daemia* [121]. In *Styphelia* species (Epacridaceae), callose walls around tetrads are very limited, resulting in fragmented exine walls [122]. In these plants, the meiotic system is likely manifested with regard to callose requirements, and so further investigation is needed to determine if callose is essential for meiotic progression in plants. Nevertheless, it may be worth noting that the differences in meiotic callose may have varying effects depending on the plant species and the specific function of callose in that species. Several apomictic plants, including aposporic *Poa nemoralis* [123], diplosporic *Elymus restisetus* [124], *Tripsacum dactyloides* [125], and tetrasporic *Lilium* and *Tulipa* species [77], have been found to lack callose walls around apomeiotic megaspore mother cells (summarized in Table 2), which may indicate, in turn, the close link of callose to meiosis progression.

## 7. Future Directions

Although the information has been fragmentary, the mechanisms driving meiosis fate decision and initiation unique to plants have been gradually clarified thanks to the efforts of many plant researchers. A genetic link between rice RNA-binding protein MEL2 and callose synthase OsGSL5 has been proposed in terms of callose accumulation and meiosis progression [73]. In fact, transcriptomic analysis of the *mel2* mutant revealed that OsGSL5 expression in premeiotic anthers is dependent on MEL2 [126]. The *Arabidopsis* mutant lacking the FST transcription factor exhibits earlier entry and asynchronous meiosis, which appears to mimic the phenotypes of *Osgsl5* mutants [73]. These findings highlight the close relationship between meiosis initiation timing, callose deposition, and strict transcriptional or translational regulation, while the underlying mechanisms remain unknown.

ROS and Ca^2+^ are known to trigger callose deposition in plant organs in many species [127], and their levels in anthers are reported to increase at the meiotic onset [96,128,129]. Thus, the mechanisms that regulate ROS and/or Ca^2+^ levels could be strong candidates for upstream drivers of MEL2 and/or GSL5 expression or activation in rice anthers. Though the role of ROS in the development of ARC-lineage somatic layers and programmed cell death of TCs has been proposed as critical [130], knowledge about its functions in SPC/PMC development and meiotic fate decision has been limited thus far. However, a mutation of rice MICROSPOROCYTELESS1 (MIL1), which is a CC-type glutaredoxin (GRXs) expressing faintly in SPCs and strongly in TCs, shows abnormally filled somatic cells instead of SPCs and PMCs in anther locules [36]. The mutants of *Arabidopsis* ROXY1 and ROXY2 and maize MALE STERILE CONVERTED ANTHER1 (MSCA1), which are orthologs of rice MIL1, also fail to form ARCs and/or SPCs [131,132]. CC-type GRXs play roles in plant response and signaling under nitrate starvation, possibly acting downstream of ROS [133]. The phenotype in those CC-type GRX mutants with defective SPC formation is similar to that of the rice *etfβ* mutant in SPC development under the nitrogen-poor condition [37]. Thus, it is highly likely that the nitrogen-auxotrophic nature of ARC/SPC development is complicatedly intertwined with ROS level control for meiosis fate decision, followed by callose deposition for proper meiosis initiation.

Other environmental factors, such as temperature, can also influence the meiotic cell cycle, causing a delay or complete loss of certain specific meiotic stages [134,135]. *Arabidopsis* plants exposed to high temperatures accelerated early prophase progression, with pachytene being an exception where it was delayed, possibly due to an impediment in crossover maturation steps [134]. In agriculturally important crops, temperature stresses on young spikelets during meiosis and/or microspore stages often cause male sterility and lower yields due to failures in PMC, microspore, and TC development [43,136]. Thus, all of the studies presented in this review will not only shed light on the missing link in machinery for mitosis–meiosis fate decisions but will also provide clues for breeding agricultural crops that are more tolerant to environmental stresses.

## Figures and Tables

**Figure 1 plants-12-01936-f001:**
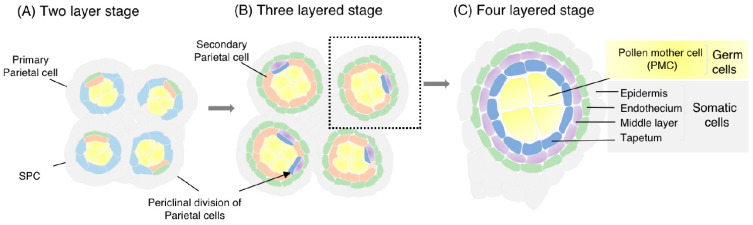
Illustration of anther differentiation and development in plants. (**A**). Cross section of the anther with two-layered somatic walls of epidermis and primary parietal cell layer (two-layered stage). Sporogenous cells (SPCs) and primary parietal cells are produced from archesporial cells in the hypodermis. The primary parietal cells periclinally divide and produce secondary parietal cells inside and endothecium outside, while SPCs proliferate mitotically. (**B**). The anther at its three-layered stage. Secondary parietal cells periclinally divide and produce a tapetum inside and a middle layer outside. (**C**). The anther is comprised of four concentric somatic layers: the epidermis, endothecium, middle layer, and tapetum from inside to out, respectively, as well as pollen mother cells (PMCs), developed from SPCs, at the central locule.

**Figure 2 plants-12-01936-f002:**
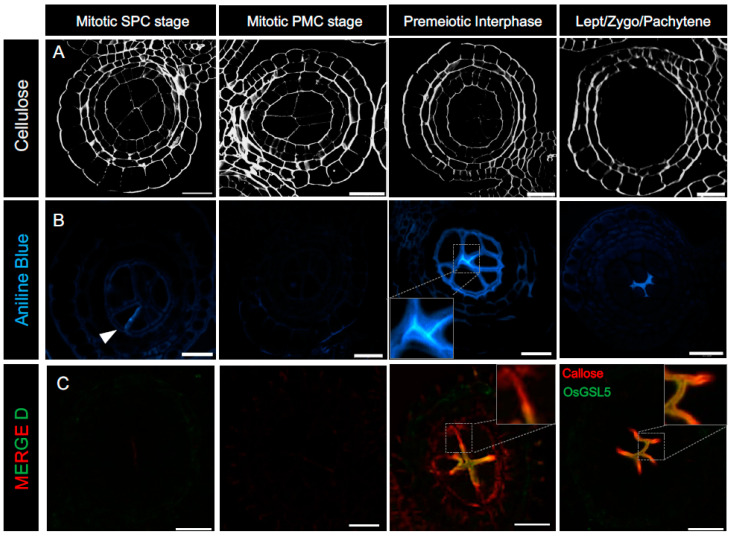
Turnover of cellulose and callose walls during the mitosis-to-meiosis transition period in rice anthers. (**A**). Staining of cellulosic walls of anther sections with Renaissance 2200. (**B**). Staining of callose walls with aniline blue. During the mitotic SPC stage, callose is not detected in anther cells, except on the newly formed cell plate between dividing daughter cells (an arrowhead). Cellulose-composing anther-locular cell walls gradually disappear alongside callose-filled intercellular spaces of anther locules around premeiotic interphase. (**C**). Immunostaining of callose (red) and OsGSL5 callose synthase (green) with specific antibodies. OsGSL5, located on the plasma membrane of PMCs, is extremely enriched at the PMC-PMC interface during the mitosis–meiosis transition phase in rice. Callose deposition is detectable at the PMC-TC interface in addition to the PMC-PMC junction, where OsGSL5 is enriched, while in subsequent early meiosis stages, it is limited to the locular center and largely corresponds to OsGSL5 localization. Scale Bar = 20 μm.

**Figure 3 plants-12-01936-f003:**
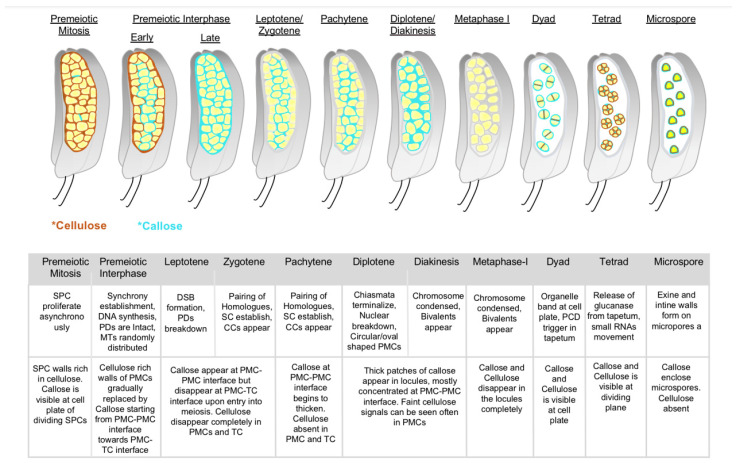
Callose deposition along with anther development from premeiosis to microspore stages. Cartoons depict callose (blue) and cellulose (brown) accumulation in pre-meiotic, meiotic, and post-meiotic anther locules as in microsporangium, based on the observations by Somashekar et al. [73] Overlap of callose and cellulose can be seen in dividing cell plates (Dyad and Tetrad). The below table shows the key meiotic and post-meiotic events taking place at each stage (upper) in correspondence to callose and cellulose remodeling shown in A (bottom). SPC—sprorogenous cells; PMC—pollen mother cells; TC—tapetal cell; PD—plasmodesmata; MT—microtubule; DSB—DNA double strand break; SC—synaptonemal complex; CC—cytomictic channels; PCD—programmed cell death. Colored asterisks indicate callose and cellulose in corresponding colors shown in cartoon.

**Figure 4 plants-12-01936-f004:**
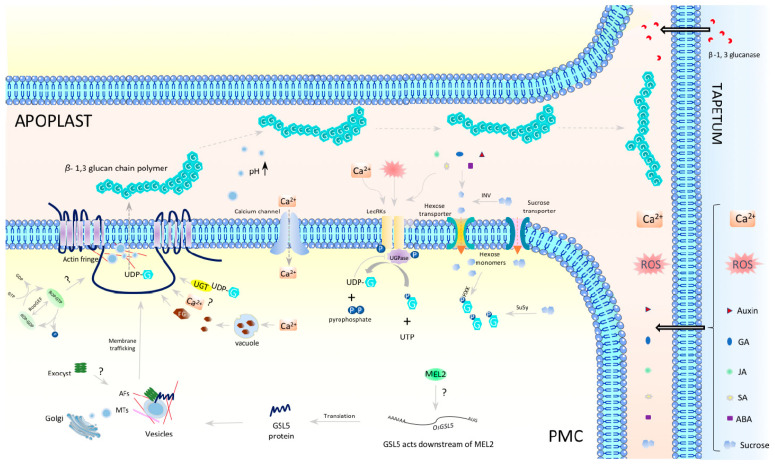
A presumable model representing regulatory mechanisms of callose biosynthesis and deposition in anther locules during mitosis–meiosis transition phase. INV—invertase; HXK—hexose kinase; LecRK—lectin rich receptor kinases; CDK—cell cycle dependent kinase; SuSy—sucrose synthase; UDP—uridine diphosphate; UTP—uridine triphosphate; UGPase—UDP-G = UDP-glucose; FG—β-Furfuryl-β-glucoside; AF—actin filament; MT—microtubule; GA—gibberellic acid; JA—jasmonic acid; SA—salicylic acid; ABA—abscisic acid; PMC—pollen mother cell; TC—tapetal cell.

**Figure 5 plants-12-01936-f005:**
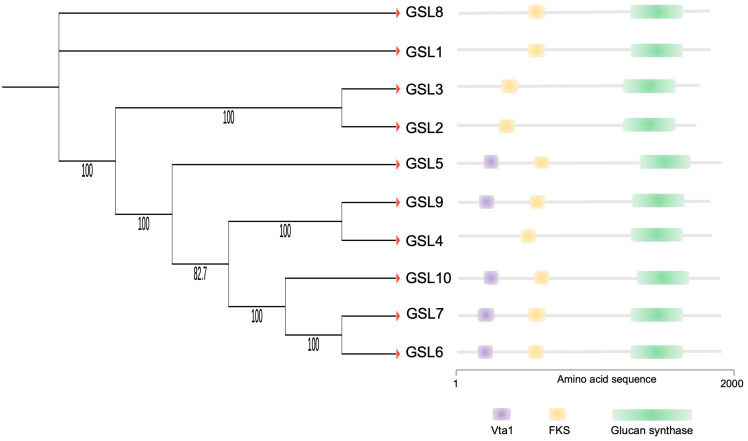
Phylogenetic relationship and conserved domains of rice GSL family proteins. Numbers associated with branches show bootstrap values. The right panel shows a schematic representation of conserved domains deduced from the amino acid sequence for each rice GSL protein.

**Figure 6 plants-12-01936-f006:**
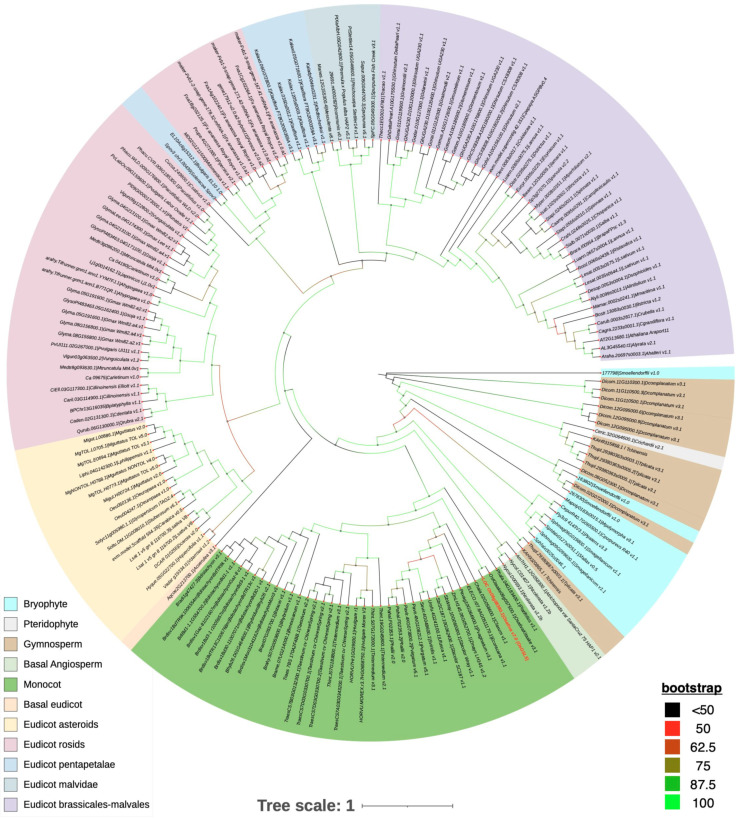
Phylogenetic tree of OsGSL5 homologues across the plant kingdom. The protein sequence of OsGSL5 was obtained from RAPDB and blasted as a query sequence to obtain its homologues across diverse plant groups on phytozome and NCBI (for gymnosperm homologues) blast search tools with the lowest E-value threshold (1 × 10^−4^). A total of 417 protein sequences were extracted and fed into the NCBI conserved domain search tool with the conserved domain database (CDD) search parameter that imports domain and protein family alignment models from Pfam, SMART, COG, PRK, and TIGRFAMs, in addition to including NCBI-curated domains and data. An e-value threshold of 1 × 10^−4^ was used for the analysis. Only the sequences with full domain composition (see main text) were selected for phylogenetic analyses. For protein sequences obtained from Phytozome and NCBI-PSI-Blast searches and NCBI conserved domain composition information, we advise readers to see Appendix A. Sequences were checked for redundancy, and duplicates were removed by the CD-HIT [116] tool. Multiple sequence alignment is carried out using MAFFT [117]. Aligned sequences were then trimmed using the trimAl [118] tool with a gap threshold of 0.999. A phylogenetic tree was constructed by IQ Tree 2.2.0 by the maximum likelihood method with an ultrafast bootstrap (1000) and SH-aLRT reliability tests, in addition to fast branch reliability evaluation [119]. To simplify the phylogenetic inference, the tree was constructed using 165 protein sequences, each representing a diverse plant species, on the interactive Tree of Life (iTOL) web interface. The phylogenetic tree was interpreted manually.

**Table 1 plants-12-01936-t001:** Genes involved in the regulation of meiosis entry, synchrony and timing in plants (Genes are shown in the order of their functionality).

Gene	Plant	Protein Type	Meiosis Progression	Expression Site/Stage	Refs.
*MIL1*	Rice	Plant-specific CC-type glutaredoxin	Sporogenous cells fail to enter into meiosis	Sporogenous cells and inner parietal cells	[36]
*AM1*	Maize	Coiled-Coiled domain protein	Failure in mitosis-meiosis switch and prophase I progression in male and female	Male meiocytes. Localize on chromatin and pericentromeres	[57,58]
*SWI/DYAD*	Arabidopsis	WINGS-APART LIKE (WAPL) inhibitor	Fail to switch female meiosis. Male meiosis is initiated but abnormal	Male and female meiocytes at G1/S phase and prophase I	[59,60]
*MEL2*	Rice	RNA binding protein	Asynchronous meiosis initiation and germ cells arrest at leptotene/zygotene	Male and female meiocytes at G1/S/G2 and phase	[67]
*OsRR24/LEPTO1*	Rice	Type-B response regulator	Leptotene/Zygotene transition	Early meiosis PMCs	[68]
*SDS*	Arabidopsis	Plant specific cyclin	Reduced frequency of pachytene I stage. Defects in synapses and sister chromatid separation	Male and female meiocytes	[55]
*TAM/CYCA1;2*	Arabidopsis	Cyclin A	Delayed pachytene I and longer meiosis II. Asynchrony from diplotene to tetrad	Male meiocyte at pachytene I stage	[46,53,54]
*MLH3*	Arabidopsis	DNA repair protein	Prophase I prolonged for ~25 h	Expressed in buds. Localize to chromosome axes	[70]
*DUET/MMD1*	Arabidopsis	PHD finger transcription factor	Prolonged, Metaphase I	Sporogenous and tapetum cells at premeiosis and meiosis stage	[64]
*SAP*	Arabidopsis	Putative transcription factor	Female germ cells fail to transit into meiosis II	Floral meristem, influorescence and young ovules	[71]
*FST*	Arabidopsis	bHLH transcription factor	Early entry and asynchrony in meiosis	Male meiocytes nuclear localized	[55]
*GSL5*	Rice	Callose synthase	Early entry and defects in meiosis progression	Male meiocytes at premeiosis, meiosis and pollen stage	[72,73]
*MEZ1*	Petunia	Putative transcription factor	Abnormal meiosis progression	Premeiotic anther stage	[66]
*Histone H1A*, *H1B*	Tobacco	Linker histone	Asynchronous meiosis II in male germ cells. Effect on female meiosis is unknown	Universal expression	[69]

**Table 2 plants-12-01936-t002:** Plants deficit in callose deposition during micro/megasporogenesis.

Plant	Family	Callose Deficit Stage/Organ	Meiosis Fate	Refs.
*Lilium candidum*	Liliaceae	In apomeiosis at MMC side walls and cross walls	Altered meiosis	[77]
*Lilium regale*	Liliaceae	In apomeiosis at MMC side walls and cross walls	Altered meiosis	[77]
*Tulipa* spp.	Liliaceae	In apomeiosis at MMC side walls and cross walls	Altered meiosis	[77]
*Pandanus odoratissimus*	Pandanaceae	Throughout male meiosis in anthers	Monads instead of tetrads. Microspores are normal	[120]
*Pergularia daemia*	Apocynaceae	Meiosis, tetrad and microspore stage in anthers	Defects in exine wall formation on pollen grain surface	[121]
*Styphelia* spp.	Epacridaceae	Tetrad (Very weak callose deposition is seen) in anthers	Fragmentaed exine wall formation	[122]
*Poa nemoralis*	Poaceae/Graminaeae	During apomeiosis in MMC micropylar region	Apospory due to abnormal meiosis	[123]
*Elymus restisetus*	Poaceae/Graminaeae	During apomeiosis around MMC	Death of some 2n megaspores/binucleate female gametophyte	[124]
*Tripsacum dactyloides*	Poaceae/Graminaeae	During apomeiosis around Megasporocytes	Diplospory due to failure of meiosis	[125]

## Data Availability

All data supporting this study are available within the article and Appendix A. Raw data supporting findings of this data are available upon reasonable request to the corresponding author.

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
