# Peer review of "Genetic Regulation of Mitosis–Meiosis Fate Decision in Plants: Is Callose an Oversighted Polysaccharide in These Processes?"

_plants, 2023, doi:10.3390/plants12101936_

Round 1
Reviewer 1 Report
This review compiles the main points associated with controlling the progress of meiosis. Manuscript needs to be better organized, especially for the descriptions and examples that could accompany the development of the meiotic phases, in addition to some examples that, in the same paragraph, mix meiosis and mitosis. Perhaps the most critical point in my understanding is the central concept of meiosis, which is definitely not a cyclic division. Based on this, I consider that the manuscript needs a major review, and, after that, it could be revised by someone with experience in plant physiology, given the many examples related to this area.
Abstract
Q1: Although some people use the terms "meiotic cell cycle" or just "meiotic cycle", meiosis is not actually a cyclic division. This is an important part of the life cycle, but this division starts in a meiotic mother cell and culminates in gametes. Gametes are not directed to another meiotic division. Therefore, I suggest that the authors re-evaluate the terms throughout the manuscript.
Keywords
Q2: Please, organize in an alphabetic order
Introduction
Q3, paragraph 1: Considering the sentence {Though meiosis is believed to have evolved from mitosis, it differs from mitosis in many aspects, such as programmed DNA double strand break (DSB) for- mation, meiosis-specific chromosome assembly, homolog pairing, synapsis, crossing over, and reductional chromosome segregation}, DNA double-strand breaks (DSBs) are well known for their deleterious effects, which are assiciated with mutations, translocations and even loss of genetic material. Although meiosis requires DSBs for development of recombination points at prophase I stage,, the use of DBS's term seems to be redundant in the example list pointed out by authors, like crssing over. Please, rephrase this sentence.
Q4, paragraph 2: Please, check the sentence {Multicellular organisms achieve mei- osis through greatly coordinated systems involving both cell autonomous and non-auton- omous mechanisms throughout the meiotic process.}. Please delete " throughout the meiotic process", because it sounds strange to target meiosis along meiosis.
Q5, paragraph 3: Please, put in the text the abbreviation of Primary Parietal Cells (PPC ??). In the figure, edit “Secondary PC, but observe that Secondary Parietal Cells (SPC) may be confused with sporogenous cells (SPCs) by some readers. It would be nice to resolve it.
Q6, paragraph 4: check the sentence {In the mel1 mutant, meiotic chromosome condensation seems stopped around meiotic leptotene, suggesting the meiosis fate decision early in ARC initiation stage, and also the importance of small RNA-mediated pathway in it.}. This is confusing. The use of "meiotic leptotene" leads us to think that there is leptotene in mitosis. It could just be "leptotene". Also, the part referring to the RNA was left vague. What are the small RNA-mediated pathways? Note also the term “meiotic cell cycle control” at the end.
Q7, paragraph 5: Check the italic format for “Arabidopsis”
Topic: Mitosis-meiosis transition.......
Q8, paragraph 6: This topic refers to the universe of plants and its first examples are in yeasts and mammals. That sounds inappropriate. There are also conceptual inadequacies about "meiotic cycle" . In fact, I think it's important to separate the examples about the cell cycle (mitosis) from those about meiotic cell division, or at least define when it comes to one division or the other. Note the lack of italicized format for scientific names. Please, check it.
Q9, Table 1: Table 1, just as descriptions and examples of regulators could follow the meiotic division sequence (as in Table 3), would be more organized for the reader.
Topic: Callose: a hallmark ........
Q10, paragraph 9: The authors use the term "meiotic anther". What it means? a mature anther? Would a non-meiotic anther be those immatures or post-meiotic? Are these terms usual? Please check if this is the best way to refer to these events.
Q11, paragraph 10: There is also a misuse of the concept “meiotic cycle” .
Q12, paragraph 11: There is also a misuse of the concept “meiotic cycle” .
Q13, paragraph 12: In this paragraph, the examples jump from pre-meiosis/meiosis to the tip of pollen tubes, and then back to the argument in the early meiotic environment. This makes understanding difficult.
Author Response
Q1: Although some people use the terms "meiotic cell cycle" or just "meiotic cycle", meiosis is not actually a cyclic division. This is an important part of the life cycle, but this division starts in a meiotic mother cell and culminates in gametes. Gametes are not directed to another meiotic division. Therefore, I suggest that the authors re-evaluate the terms throughout the manuscript.
Authors response:
We thanks Reviewer1 for raising a concern about the usage of term “meiotic cell cycle” and “meiotic cycle”.
Cell cycle is the name we use in biology to refer to a series of events that takes place in a cell as it grows and divide each time (https://doi.org/10.3390%2Fcells11040704/). These series of events (replication, sister chromatid separation, cytokinesis etc.,) always occur in same order in different stages, G1, S, G2 and M in both mitosis and meiosis, while sister chromatid separation is replaced with homolog separation in meiosis I, and DNA replication between meiosis I and II is skipped. It is well known that meiosis shares many cell cycle-controlling factors with mitosis, such as cyclin, cyclin dependent kinases, cohesin subunits and etc. In plants, meiosis can be turned into mitosis by means of knocking out of three genes involved in DSB formation, reductional division and skipping of 2nd DNA replication, respectively, which is called the MiMe technology (https://doi.org/10.1371/journal.pbio.1000124). It clearly indicates that meiosis is a modified version of mitotic cell division. From this viewpoint, we can say that meiosis is achieved by modifying two consecutive divisions of continuous somatic cell-division cycles, to halve the chromosome numbers. In addition, meiosis itself can be classified as a cyclic division because it consists of two rounds of G1/S/G2/M cycle. Actually, when searching the term "meiotic cell cycle" online, we can get more than 200,000 hits, clearly indicating this concept accepted broadly.
Hence, we believe the usage of term “meiotic cell cycle” is not problematic for article. However, to dispel the concerns suggested by the reviewer1, we revised a sentence including the term "meiotic cell cycle" (L11-12 in the 1st paragraph of the introduction) as follows;
(before) Thus, studies on cell cycle control and related aspects around ....
(after) Thus, studies on the meiosis-specific mode of cell cycle control (hereafter we call it "meiotic cell-cycle control") and related aspects around ....
We believe the above revision will make the concept more clearer, and lead to better readers' understanding of the terms “meiotic cell cycle” and “meiotic cycle” appearing through the text.
Keywords
Q2: Please, organize in an alphabetic order
Authors response:
Thanks for your useful comment. Issue addressed in the revised manuscript.
Introduction
Q3, paragraph 1: Considering the sentence {Though meiosis is believed to have evolved from mitosis, it differs from mitosis in many aspects, such as programmed DNA double strand break (DSB) for- mation, meiosis-specific chromosome assembly, homolog pairing, synapsis, crossing over, and reductional chromosome segregation}, DNA double-strand breaks (DSBs) are well known for their deleterious effects, which are assiciated with mutations, translocations and even loss of genetic material. Although meiosis requires DSBs for development of recombination points at prophase I stage,, the use of DBS's term seems to be redundant in the example list pointed out by authors, like crossing over. Please, rephrase this sentence.
Authors response:
Thanks for your useful comment. Issue addressed in the revised manuscript.
Q4, paragraph 2: Please, check the sentence {Multicellular organisms achieve mei- osis through greatly coordinated systems involving both cell autonomous and non-auton- omous mechanisms throughout the meiotic process.}. Please delete " throughout the meiotic process", because it sounds strange to target meiosis along meiosis.
Authors response:
Thanks for your useful comment. Issue addressed in the revised manuscript.
Q5, paragraph 3: Please, put in the text the abbreviation of Primary Parietal Cells (PPC ??). In the figure, edit “Secondary PC, but observe that Secondary Parietal Cells (SPC) may be confused with sporogenous cells (SPCs) by some readers. It would be nice to resolve it.
Authors response:
Thanks for the constructive comments. According to the reviewer1's comments, we removed the abbreviation "PC" for parietal cells in the text and figure 1 legend.
Q6, paragraph 4: check the sentence {In the mel1 mutant, meiotic chromosome condensation seems stopped around meiotic leptotene, suggesting the meiosis fate decision early in ARC initiation stage, and also the importance of small RNA-mediated pathway in it.}. This is confusing. The use of "meiotic leptotene" leads us to think that there is leptotene in mitosis. It could just be "leptotene". Also, the part referring to the RNA was left vague. What are the small RNA-mediated pathways? Note also the term “meiotic cell cycle control” at the end.
Authors response:
We thank the reviewer for this question. We rephased "meiotic leptotene" into just "leptotene". In addition, to make the relationship between Argonaute proteins and small RNAs clearer, we inserted several words into the sentence just in front of the above sentence to explain that Argonaute protein associates with small RNAs as guide RNAs. Please find our revision in the text.
The definition of the term "meiotic cell cycle" was already addressed, and then not revised here.
Q7, paragraph 5: Check the italic format for “Arabidopsis”
Authors response:
Thanks for your useful comment. Issue addressed in the revised manuscript.
Topic: Mitosis-meiosis transition.......
Q8, paragraph 6: This topic refers to the universe of plants and its first examples are in yeasts and mammals. That sounds inappropriate. There are also conceptual inadequacies about "meiotic cycle" . In fact, I think it's important to separate the examples about the cell cycle (mitosis) from those about meiotic cell division, or at least define when it comes to one division or the other. Note the lack of italicized format for scientific names. Please, check it.
Authors response:
The conceptual issue in the term "meiotic cycle" was already solved as mentioned in the answer to Q1. Here we aimed to drive the cell-cycle people working on other model systems to our topic i.e “Mitosis-meiosis transition in plants”, so we thought it would be necessary to shortly summarize about cell-cycle control in other famous model systems such as yeast and mammals in 1 or 2 sentences in the beginning before telling plant-specific issues. In addition, such sentences helps us to emphasize that CDKs/cyclin complexes are one common factors involved in cell-cycle control of both mitosis and meiosis in various organisms like yeasts, mammals, plants and so on, (although the mechanism is organism-specific). Perhaps, this is important for readers to understand the mitosis-meiosis transition in plants from broader perspective. We will be happy if the reviewer understand our concept here.
Q9, Table 1: Table 1, just as descriptions and examples of regulators could follow the meiotic division sequence (as in Table 3), would be more organized for the reader.
Authors response:
Thanks for your useful comment. Issue addressed in the revised manuscript.
Topic: Callose: a hallmark ........
Q10, paragraph 9: The authors use the term "meiotic anther". What it means? a mature anther? Would a non-meiotic anther be those immatures or post-meiotic? Are these terms usual? Please check if this is the best way to refer to these events.
Authors response:
Thanks for your helpful comments. We rephrased "meiotic anther" and "premeiotic anther" to "the anther undergoing premeiosis and meiosis", and added the bracketed definition for the term "meiotic anther" and "premeiotic anther" appearing in the text below. We would be happy if the reviewer finds benefits in our revision.
Q11, paragraph 10: There is also a misuse of the concept “meiotic cycle” .
Authors response:
We have addressed this issue in Q1
Q12, paragraph 11: There is also a misuse of the concept “meiotic cycle” .
Authors response:
We have addressed this issue in Q1
Q13, paragraph 12: In this paragraph, the examples jump from pre-meiosis/meiosis to the tip of pollen tubes, and then back to the argument in the early meiotic environment. This makes understanding difficult.
Authors response:
We thank the reviewer for this question. As the reviewer pointed out, the readers may be confused if the topic suddenly jumps from meiosis to pollen tube elongation. So the sentences that the reviewer1 pointed out were revised to weaken the impression that the topic has taken a leap forward to pollen tube elongation, and to make the connection between the two topics easier to understand. Please find our revision in the text.
Reviewer 2 Report
This manuscript by Harsha Somashekar and Ken-Ichi Nonomura reviewed the genetic regulation of meiosis fate decision and initiation in plants with emphasis on the functions of callose in these processes. They introduced the divergence of meiosis initiation among species and the differentiation process of anther in flowering plants. Then, studies on the molecular players in meiosis initiation and cell cycle control in plants were summarized. The working model of the biological function and deposition pattern of callose during plant meiosis, combined with the recently published work on OsGSL5 by this group, was proposed and future directions on this field was discussed.
Basically, this paper presents an accurate summary of research process on plant meiosis initiation and may provide references for further studies on this important biological processes. However, I have some concerns which are listed below to improve the manuscript.
1. The authors mentioned two processes as “meiotic fate decision” and “mitosis-meiosis transition” in two separated subtitles. How to define these two processes? Can give a more accurate definition of these two processes?
2. Figure 3, Why the PMCs at metaphase I in cartoons are free of callose? Callose is distributed around the PMCs at this stage.
3. Some minor mistakes should be checked and modified. For example, in figure 1, “Secondary PC” was wrongly presented (Figure 1B)
This manuscript by Harsha Somashekar and Ken-Ichi Nonomura reviewed the genetic regulation of meiosis fate decision and initiation in plants with emphasis on the functions of callose in these processes. They introduced the divergence of meiosis initiation among species and the differentiation process of anther in flowering plants. Then, studies on the molecular players in meiosis initiation and cell cycle control in plants were summarized. The working model of the biological function and deposition pattern of callose during plant meiosis, combined with the recently published work on OsGSL5 by this group, was proposed and future directions on this field was discussed.
Basically, this paper presents an accurate summary of research process on plant meiosis initiation and may provide references for further studies on this important biological processes. However, I have some concerns which are listed below to improve the manuscript.
1. The authors mentioned two processes as “meiotic fate decision” and “mitosis-meiosis transition” in two separated subtitles. How to define these two processes? Can give a more accurate definition of these two processes?
2. Figure 3, Why the PMCs at metaphase I in cartoons are free of callose? Callose is distributed around the PMCs at this stage.
3. Some minor mistakes should be checked and modified. For example, in figure 1, “Secondary PC” was wrongly presented (Figure 1B)
Author Response
Reviewer 2
Comments and Suggestions for Authors
This manuscript by Harsha Somashekar and Ken-Ichi Nonomura reviewed the genetic regulation of meiosis fate decision and initiation in plants with emphasis on the functions of callose in these processes. They introduced the divergence of meiosis initiation among species and the differentiation process of anther in flowering plants. Then, studies on the molecular players in meiosis initiation and cell cycle control in plants were summarized. The working model of the biological function and deposition pattern of callose during plant meiosis, combined with the recently published work on OsGSL5 by this group, was proposed and future directions on this field was discussed.
Basically, this paper presents an accurate summary of research process on plant meiosis initiation and may provide references for further studies on this important biological processes. However, I have some concerns which are listed below to improve the manuscript.
- The authors mentioned two processes as “meiotic fate decision” and “mitosis-meiosis transition” in two separated subtitles. How to define these two processes? Can give a more accurate definition of these two processes?
Authors response:
We really appreciate for your commenting the essential point. By “meiotic fate decision”, we refer to the steps from germ cell differentiation to till the SPCs mature into PMCs and acquire the meiosis fate decision but have not initiated meiosis yet. In mitosis-meiosis transition, we referred to the events that comes after PMCs acquire meiotic fate decision and mostly concerned with PMCs preparing at pre-meiotic interphase stage to initiate meiosis.
However, these we totally agree with you that they are difficult to separate with each other. So we decided to use "mitosis-meiosis fate decision", replaced to “meiotic fate decision” and "meiosis-meiosis transition". This term is also accepted in meiosis research field generally (https://cshperspectives.cshlp.org/content/3/8/a002683.full.pdf). I hope the reviewer find benefits in this revision.
- Figure 3, Why the PMCs at metaphase I in cartoons are free of callose? Callose is distributed around the PMCs at this stage.
Authors response:
In our observation in the GSL5 paper published in Plant Physiology (2023), we have checked the meiosis stage in parallel with aniline blue staining. We have not observed the callose distributing around PMCs at the metaphase-I stage. This result was summarized in the table included in Figure 3. In addition, to clarify the origin of this information more clearly, we cited Somashekar et al. (2023) in the legend of Fig. 3.
- Some minor mistakes should be checked and modified. For example, in figure 1, “Secondary PC” was wrongly presented (Figure 1B)
Authors response:
Thanks a lot for your useful comment. It may be raised by secondary PC (parietal cell) mixed up with SPC (sporogenous cell), as the reviewer 1 pointed out too. To avoid confusion for the readers, we abolished the abbreviation "PC" and replaced it to the original "parietal cell" in the Figure 1A and B and the legend.
Round 2
Reviewer 1 Report
Dear authors and editor.
I have evaluated the answers and the manuscript, and the changes have been sufficient. However, I have a very different view of the authors regarding cell divisions. I absolutely agree that is important for readers to understand the mitosis-meiosis transition in plants considering a broader perspective, therefore, I believe it would be important to take another look at what a cyclical division is. Even that many of the proteins and checkpoints are common to both divisions, and even to exist seen dozens and dozens of articles using "meotic cycle" term, the meiosis is not cyclic because the end products (gametes) don't go into new cell cycle containing the same characteristics in the premeiotic cells. Also, I don't agree with the idea of cycle, because the first and second phases are not equal. As examples I can list: M1) homologous pairing, synaptonemal complex formation, crossing-over, sharing a kinetochore structure for the two sister chromatids and reductional separation, while in M2 there are: sister chromatid reorientation, kinetochore independence in each chromatid and equational separation. These events do not refer to something cyclical, although they are dependent. Nuclei in tetrads or young pollen grains will undergo pollen mitosis with half the complement of the originating somatic cells and gametes will to move towards fertilization. In my honest opinion, I see no problem if authors explaining in a paragraph the relationships between meiosis and mitosis and using the terms "mitotic division" and "meiotic division", or simply mitosis and meiosis. This does not harm this important revision work and eliminates the idea that meiosis is cyclic. Authors could take advantage of this review to break down this concept and instigate this discussion. But, I prefer to leave that decision to the editor.